# Simulation of Starch Gel Printing and Deformation Process Using COMSOL

**DOI:** 10.3390/foods13060881

**Published:** 2024-03-14

**Authors:** Zhou Qin, Zhihua Li, Xiaobo Zou, Ziang Guo, Siwen Wang, Zhiyang Chen

**Affiliations:** 1Agricultural Product Processing and Storage Lab, School of Food and Biological Engineering, Jiangsu University, Zhenjiang 212013, China; 19599960979@163.com (Z.Q.); lizh@ujs.edu.cn (Z.L.); 2112018015@stmail.ujs.edu.cn (Z.G.); 13361224526@163.com (S.W.); 15370988685@163.com (Z.C.); 2International Joint Research Laboratory of Intelligent Agriculture and Agro-Products Processing, Jiangsu Education Department, Zhenjiang 212013, China

**Keywords:** 3D printing, starch gel, simulation, deformation, high temperature, warpage

## Abstract

The food industry holds immense promise for 3D printing technology. Current research focuses mainly on optimizing food material composition, molding characteristics, and printing parameters. However, there is a notable lack of comprehensive studies on the shape changes of food products, especially in modeling and simulating deformations. This study addresses this gap by conducting a detailed simulation of the starch gel printing and deformation process using COMSOL Multiphysics 6.2 software. Additive manufacturing (AM) technology is widely acclaimed for its user-friendly operation and cost-effectiveness. The 3D printing process may lead to changes in part dimensions and mechanical properties, attributable to the accumulation of residual stresses. Studies require a significant amount of time and effort to discover the optimal composition of the printed material and the most effective deformed 3D structure. There is a risk of failure, which can lead to wasted resources and research delays. To tackle this issue, this study thoroughly analyzes the physical properties of the gel material through COMSOL Multiphysics 6.2 software, It simulates the heat distribution during the 3D printing process, providing important insights into how materials melt and solidify. Three-part models with varying aspect ratios were meticulously designed to explore shape changes during both the printing process and exposure to an 80 °C environment, employing NMR and rheological characterization. Using the generalized Maxwell model for material simulation in COMSOL Multiphysics, the study predicted stress and deformation of the parts by analyzing solid heat transfer and solid mechanics physical fields. Simulation results showed that among three models utilizing a gel-PET plastic membrane bilayer structure, Model No. 1, with the largest aspect ratio, exhibited the most favorable deformation under an 80 °C baking environment. It displayed uniform bending in the transverse direction without significant excess warpage in the edge direction. In contrast, Models No. 2 and No. 3 showed varying degrees of excess warpage at the edges, with Model No. 3 exhibiting a more pronounced warpage. These findings closely aligned with the actual printing outcomes.

## 1. Introduction

Additive manufacturing (AM) is a technology that employs various materials to construct parts layer by layer. Fused Deposition Modeling (FDM) is a 3D printing technology wherein material is melted through an extrusion nozzle to form parts [1]. Since the introduction of 3D printing technology, its applications have become widespread in a number of fields [2], including surgical and biomedical applications, among others [1]. Three-dimensional food printing, an innovative and sustainable food manufacturing process [3] that emulates the texture of food products through the sequential stacking of pre-prepared food inks, holds significant promise within the realm of food technology [4]. FDM is renowned for its operational simplicity and cost-effectiveness. Essentially, it functions as an extrusion system enabling the fabrication of parts characterized by complex structures and intricate details. The dimensions of the part may change during the printing process, and the excessive generation of residual stresses during 3D printing can compromise the structural integrity and mechanical properties of the part, making it susceptible to cracking or collapse. At the same time, there are significant deficiencies in the modeling and simulation of deformation prediction for 3D-printed food products [5]. Controlling deformations in these printed foods is difficult due to the limited predictive data available. A lack of predictive data can lead to failed experiments and waste valuable resources, including time and materials. Therefore, a method that can obtain the condition parameters for 3D printing at a low cost and simulate the changes in the printed part while recording the data of the part changes is of great significance. This study will attempt to find such a method and provide new ideas for subsequent related research on 3D printing.

One of 4D printing’s strengths lies in its material adaptability [6], allowing the customization of material responses to specific configurations [7]. The shape of food products has an impact on consumer preferences [8], and attractive and visually striking shapes can enhance the allure and increase the perceived value of a product [9]. Four-dimensional printing is an emerging technology in the field of additive manufacturing technology that introduces the concept of changing print configurations [10], including shape and color, over time [11], thus expanding the traditional capabilities of 3D printing [7]. The concept of 4D printing was originally proposed by researchers at the Massachusetts Institute of Technology (MIT) in 2014 [12]. While there are similarities between 4D printing and 3D printing in the creation of 3D model designs and the use of 3D printer equipment [13], the main differences are in the selection of smart materials and the creation of smart models or structures. In 4D printing, these choices can control changes in the color [14] and shape of the printed object [15]. Four-dimensional printing shows great potential for development in the food industry [16]. Contemporary investigations into 4D food printing predominantly center on refining the composition of food materials, molding characteristics, and printing parameters [17]. However, there is a notable gap in harnessing the distinct advantages of food shapes to elevate product presentation [18]. Research within 4D printing, which examines the morphological transformations of food products, frequently emphasizes the deformation analysis of single-material printed models. For instance, an investigation delves into the microwave-induced spontaneous deformation of purple potato puree and grease gel [19]. There have also been some studies on the use of heated dehydration-induced deformation of 4D-printed edible insects [20]. However, these studies require a lot of time and effort to discover the optimal composition of the printed material and the most efficient 3D structure for deformation [21]. There is a risk of failure, which may lead to a degree of wasted resources and research delays. Courter et al. [22] investigated the impact of residual stresses in the simulation, considering material properties such as specific heat capacity, which were treated as temperature-dependent variables. For ease of analysis, a fixed temperature boundary condition was imposed on the print bed for the component.

COMSOL Multiphysics is a robust simulation software package [23] that plays a vital role in the analysis and optimization of various facets of the 3D printing process. It can simulate the thermal distribution during 3D printing, providing essential insights into how materials melt and solidify [24], ensuring that the temperature curve is suitable for specific 3D printing technologies and materials. Additionally, it allows for the modeling and analysis of the structural integrity of 3D-printed objects, predicting stresses, strains, and deformations during and after the printing process to optimize designs for enhanced strength and durability.

For techniques such as binder jetting or material extrusion, where fluids play a vital role, COMSOL can simulate fluid flow, which is crucial for understanding how materials are deposited layer by layer and how different parameters impact the final product. The software excels at modeling material deposition and solidification, which are essential for understanding how print layers form and ensuring that the final product meets design specifications.

COMSOL excels at simulating coupled physical phenomena, allowing for a comprehensive understanding of the 3D printing process by simultaneously modeling interactions between thermal, structural, and fluid flow aspects. Through parametric studies, COMSOL can optimize various process parameters, including temperature, speed, layer thickness, and material properties, to achieve desired outcomes such as defect reduction, improved surface finish, and enhanced mechanical properties.

Supporting virtual prototyping, COMSOL enables testing and iterative design in a virtual environment before prototypes are physically manufactured. This capability saves time and resources in the product development cycle. By analyzing and predicting factors such as stress and deformation in printed components, COMSOL helps improve print quality and process efficiency. We believe that COMSOL is capable of meticulously examining the physical properties of gel materials and predicting material changes throughout the entire printing process. Utilizing COMSOL for pre-experimental validation of complex model structures allows for the verification of modes and scenarios before committing substantial human and time resources to formal experiments. By using COMSOL for the simulation of printing parameters, it is possible to efficiently generate the necessary data for the optimization of these parameters. The essence of 4D materials resides in forecasting the morphological alterations of the material in response to external stimuli [25]. Utilizing COMSOL modeling and simulation for the 3D printing process is acknowledged as an effective approach for understanding the interplay between shrinkage, warpage, and printing conditions. Nonetheless, there exists a scarcity of research focusing on modeling the shrinkage and warpage of 3D printed components. Furthermore, the impacts of different printing conditions on residual stresses and part deformation remain insufficiently explored. Through COMSOL’s simulation, the deformation and behavior of materials during food printing can be predicted more accurately, which helps to optimize printing parameters and design. Secondly, COMSOL can help researchers gain insights into the heat transfer, rheology, and curing process of food materials in high-temperature environments, which can provide an important reference for the development of more effective food printing processes. In addition, simulation using COMSOL can also reduce the cost and time of experiments and accelerate the development of food 3D printing technology. Therefore, the application of COMSOL to material simulation in food 3D printing is not only necessary but also innovative and is expected to promote the further development of food 3D printing technology.

In this study, we used starch and gluten to make printables. Starch is a commonly used ingredient and is popular in daily food as an important source of sugary nutrients. The mechanical properties of starch gels vary with different straight-to-branched ratios, so it is possible to adjust the straight-to-branched ratio of starch to obtain the desired starch gel material. Gluten is rich in protein, which is an important nutrient needed by the human body, so it is used as a protein additive. We used COMSOL to construct two physical fields, solid heat transfer and solid mechanics, and performed simulation experiments of high-temperature dehydration-induced deformation on three models of the same length but different widths. Subsequently, we conducted actual experiments to validate the simulation results and performed an in-depth analysis of the model deformation as well as stress-induced warpage. By investigating the effects of stress and model shape differences on the printing results and the effects of induced warping, we laid the foundation for further research on making more complex models from food materials. This study is expected to provide valuable insights into understanding and optimizing phenomena such as shrinkage and warpage during 3D printing.

## 2. Materials and Methods

### 2.1. Material Selection

The physicochemical properties and rheological characteristics of starch gels are directly and significantly affected by the ratio of straight-chain starch to branched-chain starch (straight-to-branched ratio), protein content, and the starch-to-water ratio of the starch itself [26], which in turn directly affects the effect of 3D printing.

The materials used in the experiments included the following: Baizuan medium gluten flour (with an amylose-to-amylopectin ratio of 0.3 and a protein content of 12%), purchased from RT-Mart Supermarket, Zhenjiang, China; high straight-chain starch (with an amylose-to-amylopectin ratio of 1.8) and gluten (with a protein content of 75%), purchased from Henan Shengfa Bio-Technology Co., Ltd. in Henan, China; and qualitative filter paper and ethanol, purchased from Zhenjiang Huadong Ware Chemical Glass Co (Zhenjiang, China).

#### 2.1.1. Preparation of Starch Gel

The physicochemical properties, rheological properties, and 3D printing characteristics of starch gels are significantly affected by their water content, straight-chain ratio, and protein content, which are the main influencing factors. The straight branching ratio and protein content can be adjusted by adding high straight-chain starch and gluten to the starch mixture. The relevant formulae are given below:$$\frac{0.3x}{1.3}+\frac{1.8y}{2.8}=100\alpha /(1+\alpha )$$
12x+75z=100β
$$\frac{100}{100+w}=100$$
x+y+z=100

In each formulation, the mass of solids was 100 g, where *x*, *y*, *z*, and *w* represented the mass (g) of medium gluten flour, high straight-chain starch, gluten, and water, respectively; and *α*, *β*, and *γ* denoted the straight-branching ratio of starch gel, protein content (%), and gel water content (%), respectively.

Baizuan medium gluten flour, obtained from RT-Mart supermarket in Zhenjiang, China, was combined with high straight-chain starch and gluten, sourced from Henan Shengfa Bio-technology Co., Ltd. in Zhengzhou, China. These ingredients were meticulously weighed in varying proportions and thoroughly blended in a beaker. Subsequently, the mixture was gradually introduced into distilled water at 50 °C, ensuring that the water constituted 75% of the total weight of the mixture, and was well stirred using a stirrer.

As the water bath was heated to boiling, the beaker was placed in the water bath. Continuous heating of the water bath ensured that the mixture rapidly warmed to 95 °C in the beaker. Throughout the process, the stirrer was maintained at 150 r/min to ensure that the starch was fully pasted. Subsequently, the beaker was quickly placed in cold water to rapidly cool the mixture to 50 °C. The mouths of the cups were sealed with plastic wrap. Subsequently, the beakers were transferred to a refrigerator set at 4 °C and allowed to stand for 24 h to facilitate the aging process of the starch, leading to the formation of a starch gel Figure 1 shows a schematic diagram of a starch gel model and its printed physical object.

#### 2.1.2. Printing Parameter Settings

The model of the sample was drawn by SolidWorks 2020, sliced, and printed using Ulti maker Cura 4.12.0. The specific printing parameters were as follows: a nozzle speed (V) of 50 mm/s, a nozzle diameter of 0.8 mm, an ambient temperature of 25 degrees Celsius, and an infill rate set to 100% (100% infill rate was chosen because the part is more prone to shrinking/warping). The bottom layer thickness and top layer thickness were both set to 0 to ensure that each layer always printed consistently. By adjusting the wall thickness, we ensured that the model did not incorporate any infill structures and consisted entirely of shell lines.

In this study, the processing parameters for the starch gel model, including the model’s length and width, thickness, line diameter and spacing, heating temperature, and raster pattern, differ. The model dimensions and styles in COMSOL strictly adhere to the model parameters listed in Table 1, and solid heat transfer and solid mechanics physics fields are incorporated. The processing parameters for the 3 samples in the simulation are outlined in Table 2. PET plastic film and starch gel materials and their material properties are already included in the COMSOL material library. It is only necessary to enter the material properties (e.g., rheological and textural properties) of the starch gel. Therefore, it is necessary to test the rheological and textural properties of the finished starch gel material and evaluate the suitability of the material for 3D printing using the rheological and textural data.

### 2.2. Modeling Methodology

This research entails simulations in the multi-physics field, encompassing heat transfer and solid mechanics. Throughout the simulation process, the model was incorporated into the print molding process by integrating the temperature (*T*) function. To emulate the layer-by-layer deposition of starch gel, we imported the print path program file, containing the part’s geometry, into COMSOL software. This procedure gradually activates the model’s elements, adjusting them in accordance with the print path and pattern. In this investigation, material parameters (e.g., specific heat capacity *Cp*, density *ρ*, and thermal conductivity *λ*) were expressed as functions of temperature *T*. This facilitated a comprehensive assessment of the mechanical properties and deformation conditions of the printed model throughout the printing process. Following this, the acquired data underwent rigorous analysis and discussion.

Within COMSOL, element activation can be performed based on the heat source, material deposition, temperature, or time. In this study, elements were activated in accordance with the material deposition from the nozzle. This activation is achieved by importing the print path file of the internal component, thereby activating the element according to the print path. To enhance the simulation process, we implemented a constant temperature boundary condition on the bottom layer to replicate the properties of the print bed. In terms of boundary conditions, we permitted deformation and warping of the bottom layer, facilitating flexibility in detaching the model from the build platform. Notably, this methodology has been successfully employed and documented in the work of Courter et al. [22].

To emulate the thermo-viscoelastic behavior of the polymer, we have employed the generalized Maxwell (GM) model. The generalized Maxwell (GM) model serves as an expansion of the original Maxwell model, a viscoelastic model employed for characterizing the rheological properties of materials. The fundamental structure of the Maxwell model involves a spring and a dashpot arranged in series, symbolizing the elastic and viscous components of the material, respectively. In contrast, the generalized Maxwell model features numerous Maxwell elements arranged in parallel, making it especially adept at describing materials with intricate viscoelastic characteristics. In such materials, a singular relaxation time may fall short of capturing the overall behavior. Through the consideration of multiple relaxation times, the generalized Maxwell model enables a more comprehensive representation of viscoelasticity. Under the influence of external force, the representation of the generalized Maxwell model is as follows:$$\sigma \left(t\right)=\sigma_{0}\exp(-\frac{t}{\tau_{m}})$$

$\sigma_{0}$ represents the initial stress at time *t* = 0, and $\tau_{m}$ denotes the relaxation time constant. The elastic spring modulus ($G_{0}$) and the dashpot viscosity ($\eta_{0}$) in the generalized Maxwell (GM) model can be expressed as follows:$$\tau_{m}=\frac{\eta_{0}}{G_{0}}$$

The representation of the relaxation shear modulus function in a Prony series is as follows [27]:$$G\left(t\right)=G_{\infty }+\Sigma_{i=1}^{n}G_{m}\exp(-\frac{t}{\tau_{m}})$$

Here, $G_{\infty }$ denotes the modulus at infinite time, $G_{m}$ stands for the elastic modulus of the spring, $\tau_{m}$ represents the relaxation time constant of the spring-dashpot pair within the same branch, and n indicates the total number of Maxwell units in the model. The instantaneous shear modulus is defined as follows:$$G\left(t\right)=G_{\infty }+\Sigma_{i=1}^{n}G$$

### 2.3. Quantification of Bending Angle

To quantify the lateral bending of the models, the quantification of the lateral bending angles was confined to the central section of the models as a cross-section. Tangents were drawn at the ends of the cross-section, and the angle between the two perpendiculars to these tangents was considered as the bending angle of the models (Figure 2).

The morphological features of the samples were captured using a camera, and the images were imported into Rhino 7.9 software. Rhino, a robust 3D modeling software widely utilized across diverse fields including 3D animation, scientific research, and mechanical design, facilitated the annotation of bending angles using its angle measurement tools.

### 2.4. Heat Transfer Studies

Contemporary research predominantly underscores the significance of monitoring the heat transfer process to predict the FDM process accurately. Heat generation initiates upon extrusion of the print material from the printhead, with heat dissipation being a continual occurrence throughout. Consequently, the integration of temperature (*T*) into the printing process becomes imperative. In modeling, the general energy balance relationship considering heat transfer is usually expressed through the equation:$$\rho C_{p}\frac{\partial T}{\partial t}-\nabla .\left(\lambda \nabla T\right)=Q$$
where ***ρ*** is the density, $C_{p}$ is the specific heat capacity, λ is the thermal conductivity, and Q is the heat source capacity.

## 3. Results

### 3.1. Starch Gel Characterization

#### 3.1.1. Nuclear Magnetic Resonance Analysis

Since the distribution of water in the material is closely related to the structure and rheological properties of the material, we have investigated the spin–spin relaxation time of starch gel, as shown in Figure 3.

Measurements of relaxation times (T_2_) at magnetic field strengths of less than 0.5 T provide insight into the amount and migration of water in the material matrix [28]. Lower relaxation times indicate that the water in the sample has less mobility and a greater propensity to bind to matrix components [29], whereas higher relaxation times suggest the converse. There are three peaks (T_21_, T_22_, and T_23_) in the curves, representing bound water (tightly bound to the macromolecular material), fixed or semi-bound water (not tightly bound to the macromolecular material), and free water (strongly mobile), respectively (Figure 2). The relative intensity of free water is close to zero, indicating a low content of free water. This is consistent with the findings of Zheng et al. [30], probably because water molecules are encapsulated within the three-dimensional mesh structure of the gel when starch is aged to form starch gels. The samples displayed reduced partially bound water and increased bound water, suggesting a closer interaction between water molecules and starch molecules, resulting in the formation of a dense network structure.

#### 3.1.2. Rheological and Textural Properties

The apparent viscosity profile of the starch gel formulation presented in Figure 4B shows that the apparent viscosity of the starch gel gradually decreases as the shear rate increases. This indicates that the starch gel is an accelerating fluid with shear-thinning properties, making it suitable for extrusion through the print nozzle and capable of maintaining the shape of the line during the printing process. According to the viscoelastic properties of starch gels, the energy storage modulus (G’) of starch gels is higher than the loss modulus (G″) in the linear elastic region (Figure 4A,C), indicating their potential to form elastic gels or gel-like structures. The gradual increase in G’ and G″ with increasing oscillation frequency indicates an increase in the internal friction of the material, suggesting that its fluidity and shape retention are more suitable for 3D printing. The gel samples exhibit moderate hardness and elasticity in Table 3, indicating a closer interaction between the water molecules and starch molecules to form a dense network structure.

### 3.2. Simulation to Model Deformation

In this study, we initially constructed fence models with varying aspect ratios. Subsequently, we employed the Maxwell model to establish corresponding simulation units in COMSOL Multiphysics, model deformation effects due to high temperatures were simulated using COMSOL, and simulation results for three model deformation effects were produced.

This model utilizes the generalized Maxwell model in COMSOL for extrusion, and the meshing of the model is based on the size of the units deposited from the nozzle (Figure 5).

Elements within the model can be activated based on factors such as temperature, material deposition, time, or heat source, as depicted in Figure 6. After introducing the solid heat transfer physics field at 80 °C, the material properties of the three predefined aspect ratio models and the material properties of the gel PET plastic film bilayer structure are incorporated into the corresponding solid mechanics physics field. Through COMSOL calculations, it is evident that Model 1 exhibits uniform bending deformation in the transverse direction, displaying the largest deformation angle among the three models. In contrast, the bending angles in the transverse direction for models 2 and 3 sequentially decrease. Upon closer observation, it is noted that models 2 and 3 undergo some changes at the four corners, showing signs of possible corner warping. Preliminary observations suggest that the aspect ratio of the model significantly influences its deformation extent and impact. Among the three existing models, the model with a smaller aspect ratio is more prone to experiencing excess warping at the corners, negatively affecting the deformation outcome. However, further empirical experiments are crucial for validating the simulation results.

### 3.3. High-Temperature Baking Induced Deformation

This study used starch gels printed on PET films. Three identical parts were printed for each of the three models, and three sets of parallel processing were conducted to obtain the most reliable data. Models are subject to shrinkage stresses due to dehydration during baking at 80 °C. The plastic films, in contrast, do not experience dehydration stresses. In contrast, the plastic film does not experience dehydration stress, so the two shrink differently, causing the sample to bend laterally towards the side of the sample that shrinks more as a whole. For the three different models that have been modeled and physically printed, we need to analyze their deformation effects under high-temperature baking conditions. The desired deformation requires the largest and most pronounced transverse deformation, and excess warpage at the edges of the model should be the smallest and least pronounced to obtain a model with relatively optimal deformation. As the heating process proceeds, the weight of water evaporated from the three models gradually decreases (Figure 7a), the water content in the models gradually decreases, and the rate of water loss slows down. When the water content of the models dropped below 20%, the dehydration rate slowed down significantly (Figure 7b–d). This is consistent with the findings of Liu et al. [31]. Furthermore, within the initial 20 min, Model 1 exhibits the least weight of water loss per unit time, followed by Model 2, with Model 3 demonstrating the fastest rate of water loss among the three. The increased dewatering speed is likely attributed to the larger specific surface area of the model, facilitating enhanced water transfer and dissipation.

The bending angle of the various models exhibited a positive correlation with the duration of heating. As the heating process proceeded, the bending degree gradually increased, especially in the pre-heating period, and the rate of change was faster, which can be explained by the fact that the models experienced rapid dehydration in the pre-heating period. However, after the heating reached 24 min, the rate of change in the degree of bending slowed down and no more significant shape changes occurred. This may be due to the transformation of the starch gel from the “elastic state” to the “glassy state”, and the formation of the “skeleton architecture”, so that the sample no longer undergoes deformation. In addition, model No. 1 (Figure 8a) shows a greater degree of deformation and bending than model No. 2 (Figure 8b) and No. 3 (Figure 8c), which is consistent with the results simulated in COMSOL.

During the deformation induced by baking, the edges of the three models exhibited varying degrees of excess warping. Once the models have stabilized, the naked eye can observe that the bending angle of Model 1 in the transverse direction is larger than that of models 2 and 3. Additionally, there is noticeable excess warping at the corners of models 2 and 3. The deformation effects of the actual model are generally consistent with those described in 3.2 Simulation to model deformation, again with Model 1 having the most pronounced lateral bending effect and the least amount of edge warping, aligning closely with the results simulated in COMSOL (Figure 6).

In this study, all three models exhibited varying degrees of excess warping at their edges. Following a discussion and a review of the pertinent literature, it was initially hypothesized that the thermal effects within the models caused differential heating between their interiors and exteriors during the heating process. Additionally, the adjacent deposition layers at the models’ corners were influenced by thermal inertia, resulting in temperature differentials between their interiors and exteriors throughout the heating process. This differential heating leads to uneven gel contraction due to varying water loss in different parts of the models, ultimately resulting in the observed excess warping at the edges of the models. Additionally, due to the wider width of Model 2 and Model 3, the differences in the shrinkage ratio within the deposition layers are not sufficient to cause extensive warping. It mainly affects the regions near the corners, with minimal impact on the central section, where bending deformation occurs primarily due to the differences in shrinkage ratio between the gel-PET plastic film bilayer structure. This phenomenon results in the generation of excess warping, attributed to anisotropic shrinkage changes induced by built-in residual stresses [32]. Furthermore, among the three models in this study, the wider the width, the more pronounced the occurrence of excess warping.

The change in bending angle under baking deformation at 80 °C was measured and compared for the three models using the method mentioned in Section 2.3, followed by a comprehensive analysis. During the heating and dehydration process, weight measurements and bending angles were recorded periodically. The morphological features of the samples were captured using a camera, and the images were imported into Rhino 7.9 software. This enabled the quantification and documentation of the models’ bending variations during the baking process (Figure 9).

The bending angles of the three models exhibited a positive correlation with heating time in the early stages. The bending angles gradually increased during the heating process, reaching a maximum value. However, in the later stages, there was a tendency for the bending angles to decrease with time. The initial rapid increase was attributed to the models undergoing rapid dehydration, with water loss becoming progressively more challenging as the water content in the models decreased. As the models approached their maximum bending angles, the rate of change in bending decreased, and noticeable shape variations ceased. This trend aligned with the findings of Liu et al.’s study [33], possibly attributed to the transition of starch gel from an “elastic state” to a “glassy state,” and the formation of a “skeletal structure” preventing further deformation [31].

Notably, Model 1 exhibited the largest lateral bending angle among the three models and was the first to initiate bending. Model 1’s bending angles reached a significant scale before models 2 and 3 began to exhibit bending angles. Additionally, Model 1 took longer to reach its maximum bending angle compared to models 2 and 3. This delay can be attributed to the larger surface area of Model 1, resulting in a faster dehydration rate in high temperatures and a longer time required to achieve the maximum bending angle. After reaching their maximum bending angles, all three models showed a gradual decrease in bending angles, indicating a shape recovery phenomenon possibly due to the breakdown of the “skeletal structure” in the dried samples, leading to fractures in the line structures. It is noteworthy that the PET film’s blank heating experiment ruled out the possibility of shape recovery caused by film shrinkage.

In contrast, models 2 and 3 exhibited a sequential decrease in lateral bending angles. The presence of noticeable excess corner warping in models 2 and 3 negatively affected the induced deformation, particularly at the four corners. The aspect ratio of these models significantly influenced their deformation, with smaller aspect ratios leading to more pronounced excess corner warping, adversely impacting the deformation effect.

## 4. Conclusions

The experimental results show that model 1, which has the smallest width (15 mm), exhibits the most favorable deformation during the 80 °C baking-induced deformation at a constant length (55 mm). It shows the largest uniform bending angle in the transverse direction among the three models. According to the physical field simulation in COMSOL and actual experimental verification, Model 1 exhibits the least excessive buckling after high-temperature deformation. On the other hand, Model 2 (33 mm wide) and Model 3 (57 mm wide) exhibit reduced bending angles in the transverse direction, accompanied by an increased effect of excessive warping at the corners, leading to a decrease in the induced deformation effect. In summary, among the three models, the model with a size of 56 mm by 15 mm length exhibits excellent bending deformation at 80 °C, which is highly consistent with the results of the COMSOL Multiphysics simulation, and to a certain extent proves the reliability of the simulation of induced deformation of starch gel 3D-printed products using COMSOL.

## 5. Future Work

In this project, we have realized the “activation” of a two-dimensional model of starch gel to form a three-dimensional structure under high-temperature stimulation and simulated the changes in the process of gel printing and “activation” using COMSOL. However, this was only a basic attempt to study the deformation of three-dimensional structures, and the simulation model needs to be further improved to support the study of the deformation of three-dimensional structures. In addition, subsequent research could start with excess warpage and explore ways to weaken or even eliminate the effects of warpage on parts.

## Figures and Tables

**Figure 1 foods-13-00881-f001:**
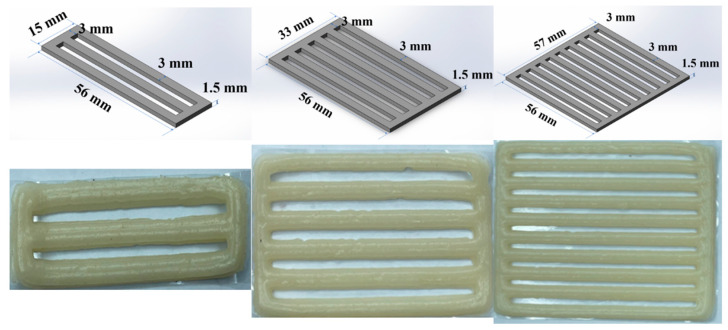
Three models and printed objects.

**Figure 2 foods-13-00881-f002:**
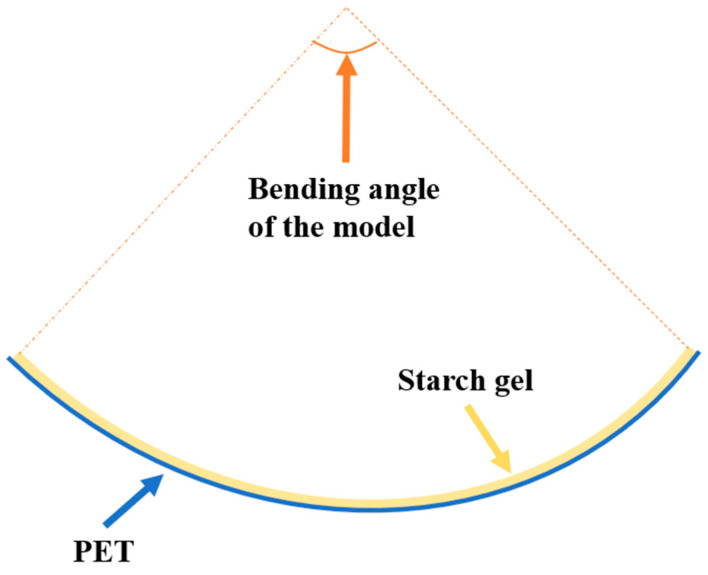
Schematic representation of the model’s bending angle quantification method.

**Figure 3 foods-13-00881-f003:**
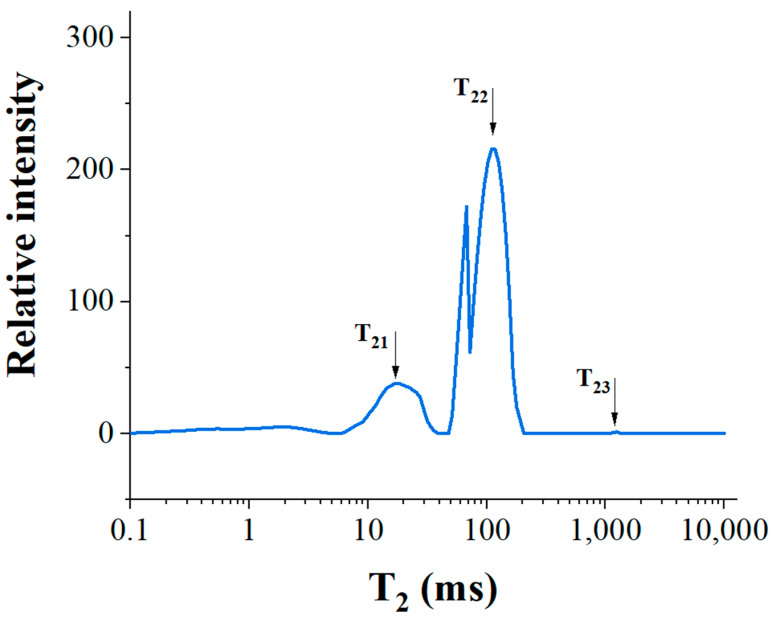
Water distribution of starch gel.

**Figure 4 foods-13-00881-f004:**
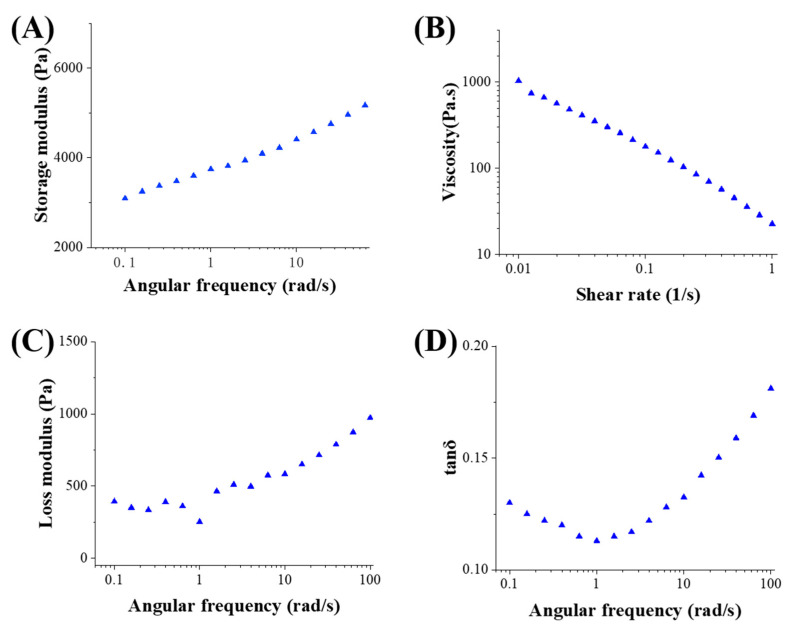
Rheological behavior: Apparent viscosity (**A**), G’ (**B**), G″ (**C**), and tan δ (**D**) of starch gel.

**Figure 5 foods-13-00881-f005:**
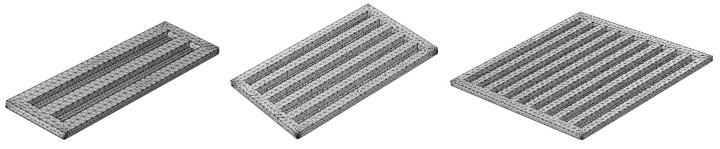
Modeling meshes in COMSOL.

**Figure 6 foods-13-00881-f006:**
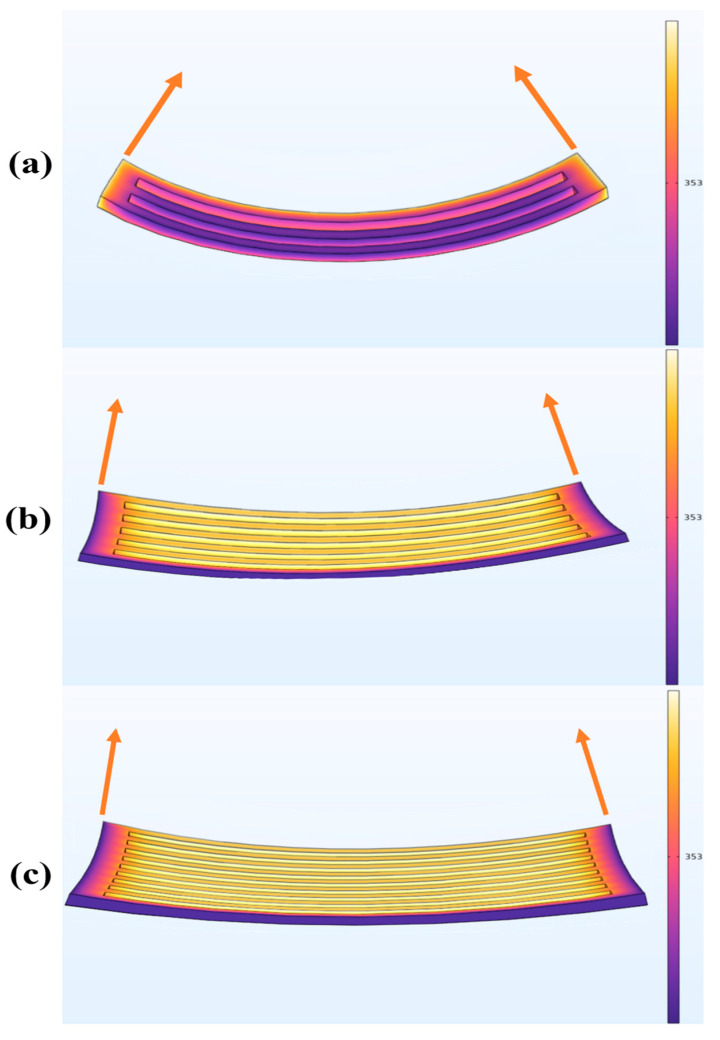
Deformation effects of three different aspect ratio samples simulated in COMSOL at 80 °C (353 K) physical field ((**a**–**c**) are the simulation results for model 1, model 2, and model 3, respectively).

**Figure 7 foods-13-00881-f007:**
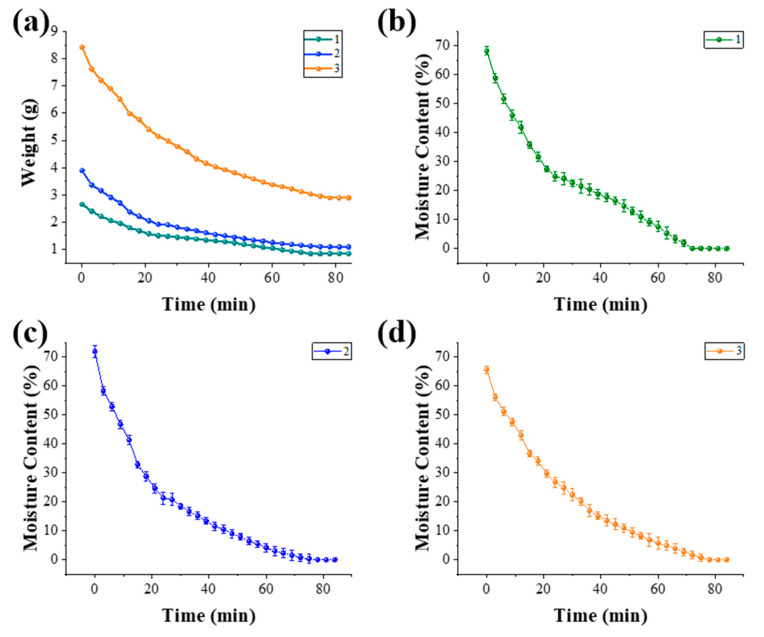
Changes in weight throughout the induced deformation for the three models (**a**) and changes in moisture content for model 1 (**b**), model 2 (**c**) and model 3 (**d**).

**Figure 8 foods-13-00881-f008:**
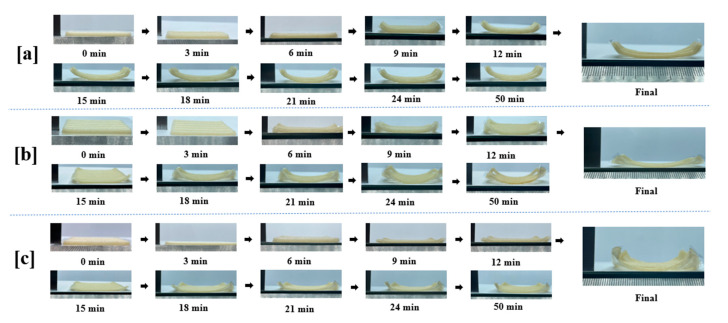
Deformation effects of model 1 (**a**), model 2 (**b**) and model 3 (**c**) with different aspect ratios during heating at 80 °C.

**Figure 9 foods-13-00881-f009:**
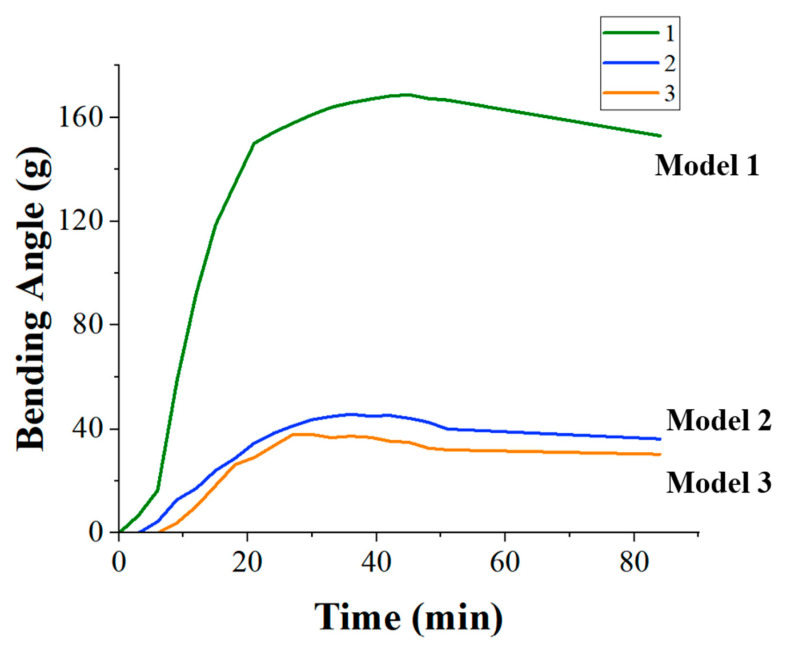
Changes in uniform bending deformation angles in the transverse direction during the baking process for the three models.

**Table 1 foods-13-00881-t001:** Dimensional parameters of the three models.

Model No.	Length	Width	Thicknesses	Line Diameter	Spacing
1	56 mm	15 mm	1.5 mm	3 mm	3 mm
2		33 mm			
3		57 mm			

**Table 2 foods-13-00881-t002:** Sample processing parameters in COMSOL.

Model No. in COMSOL	Length(mm)	Width (mm)	Thicknesses (mm)	Line Diameter (mm)	Spacing (mm)	Temperature (°C)	Raster Pattern
1	56 mm	15 mm	1.5 mm	3 mm	3 mm	80 °C	3-line fence
2		33 mm					6-line fence
3		57 mm					10-line fence

**Table 3 foods-13-00881-t003:** Texture characteristics (Hardness, Springiness, Cohesiveness, and Gumminess) of starch gels with different formulations.

Model No.	Hardness	Springiness	Cohesiveness	Gumminess
3	1543	0.885	0.743	1051

## Data Availability

The original contributions presented in the study are included in the article, further inquiries can be directed to the corresponding author.

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
