# Peer review of "Simulation of Starch Gel Printing and Deformation Process Using COMSOL"

_foods, 2024, doi:10.3390/foods13060881_

Round 1

Reviewer 1 Report

Comments and Suggestions for Authors

This paper is presenting a research on the 3D food printing. The topic is interesting as it is a part of the additive manufacturing, a very popular these days in wide range of fields.

The introduction provides good overview of the field with high stress on 4D printing, which is not mentioned in abstract (although it should). There is few places where significant editing is needed, for example in lines 50 - 54, there is hard to see what authors wanted to state. In the end sentence of the introduction authors state what this study is to provide, which I find inappropriate at that point.
In the paragraph 2.1 authors state material selection where they do not state why these materials were selected and are important from the study point. Lines 156 - 160 is repeating of the data stated in 2.1. In the experimental part there is description about the modelling, but methods and equipment used for validation are not described. In line 201 change the symbol for density.
What is meaning of x-axe T2 in Figure 2? Please give reference to the statements about water, given in L24 - 258.
First two sentences in 3.3 are stating the same. Could you please comment in the text how the smallest sample has the highest mass (fig 6a). How many samples prof one form has been printed? In sentence l320 authors state that the aspect ratio of the model highly influences the dewatering speed, but observing fig 6bcd one can see that time from roughly 70% of moisture is at all models at the beginning and they are all reaching 20% of moisture after 30+ minutes of heating with model 1 needing longest and model 2 needing shortest.
In L340 authors state that results of the physical model closely aligns with the simulated results - is there some correlation coefficient backing this statement? Unfortunately, it is hard to observe that only by comparing visually figs 5 and 7. Assessing the bending angle and comparing it to the simulation would be stronger proof. 
I suggest that the process of image modulation to get bending angles should be better described in methods part.
Figure 9 would be much easier to observe and analyse if presented in 2D.
In L393 authors state that model was printed while in the 2.1.2 there is named sample. I would suggest to make better denomination between simulated and printed samples. In L397 authors state "excellent bending" - there is nowhere defined what excellent bending would be. In the next paragraph (L399) authors conclude about the difference in heating, but there are no results supporting this statement. Similar to my statement earlier, in L422 there is also statement about excellent bending deformation.
Conclusion part is generally to long. The aim of this paper was to determine COMSOL simulation of 3D printed objects. And the results and conclusions are more stressing the samples behaviour in surrounding of elevated temperature, this should be improved. Additionally, is there enough sample/models to evaluate the system for modelling? In my opinion having only three samples with same structure (fence like) should be improved. Presented results indicate that this modelling system could be solution, but on the presented results we cannot fully agree or decline it.

Comments on the Quality of English Language

Moderate English edition should be made.

Author Response

Dear Reviewer:

Your review is very detailed and pertinent, pinpointing areas of improvement in the manuscript. I feel that you must have reviewed my manuscript very carefully and had the idea of instructing a back like me, so you pinpointed my shortcomings.

Thank you very much for your work, I have carefully considered all the comments you have made and tried to revise them in accordance with your comments, I hope you will contact me if there are any shortcomings, and I will respond as soon as possible.

Thank you again for your guidance!

Sincerely

Author

Response to Reviewer's Comments

Summary

Thank you very much for taking the time to review the manuscript. I have responded to each of your comments below and have made the appropriate revisions in the resubmitted manuscript.

Comments 1:

The introduction provides good overview of the field with high stress on 4D printing, which is not mentioned in abstract (although it should). There is few places where significant editing is needed, for example in lines 50 - 54, there is hard to see what authors wanted to state. In the end sentence of the introduction authors state what this study is to provide, which I find inappropriate at that point.

Response 1:

Thank you for pointing this out. I agree with this comment.

The language of the abstract was adjusted by adding a description of the limitations of the current study (lines 22-31).

Comments 2:

There is few places where significant editing is needed, for example in lines 50 - 54, there is hard to see what authors wanted to state. In the end sentence of the introduction authors state what this study is to provide, which I find inappropriate at that point.

Response 2:

Restructured the first paragraph of the introduction.

Firstly, the application areas of Additive Manufacturing (AM) are introduced, followed by a detailed description of the application of AM in food manufacturing and the associated features. Then the limitations of the existing research are pointed out and the measures that need to be taken to break these limitations are proposed. (lines 51-76)

Comments 3:

In the paragraph 2.1 authors state material selection where they do not state why these materials were selected and are important from the study point.

Response 3:

Thanks for the correction. The reasons for the choice of raw materials are explained in the last paragraph of the introduction. (Lines 163-168)

Comments 4:

Lines 156 - 160 is repeating of the data stated in 2.1.

Response 4:

Redundant data has been deleted.

Comments 5:

In the experimental part there is description about the modelling, but methods and equipment used for validation are not described.

Response 5:

Data on the generalized Maxwell model are already available in COMSOL, and only a brief introduction to the Maxwell model is given here. Moreover, PET plastic film and starch gel materials are already available in the material library of COMSOL, so users only need to input some material properties (e.g., rheological properties, textural properties, etc.) of the starch gel. A description has been added to 2.1.2 (lines 242-247).

Comments 6:

In line 201 change the symbol for density.

Response 6:

Modified (line 257)

Comments 7:

What is meaning of x-axe T2 in Figure 2? Please give reference to the statements about water, given in L24 - 258.

Response 7:

T2 indicates the relaxation time (already added in line 330) and a description of water has been added as a reference to support it (in revised draft)

Comments 8:

First two sentences in 3.3 are stating the same. Could you please comment in the text how the smallest sample has the highest mass (fig 6a).

Response 8:

Criteria for evaluating the best quality have been added (lines 397-401).

“The desired deformation is transverse deformation is the largest and most pronounced, and excess warpage at the edges of the model is the smallest and least pronounced.”

Comments 9:

How many samples prof one form has been printed? 

Response 9:

Three samples were obtained from each model printout, and multiple measurements yielded the Error bar (lines 391-393)

Comments 10:

In sentence l320 authors state that the aspect ratio of the model highly influences the dewatering speed but observing fig 6bcd one can see that time from roughly 70% of moisture is at all models at the beginning and they are all reaching 20% of moisture after 30+ minutes of heating with model 1 needing longest and model 2 needing shortest.

Response 10:

The rate of dewatering in the text refers to the weight of water lost per unit time, not the rate at which the percentage of water content decreases. Since the weight of water lost per unit time is faster for models with a larger surface area, the surface area has a large effect on the rate of dehydration, not the transverse to vertical ratio, which I will restate in the text (lines 402-409).

Comments 11:

In L340 authors state that results of the physical model closely aligns with the simulated results - is there some correlation coefficient backing this statement? Unfortunately, it is hard to observe that only by comparing visually figs 5 and 7. Assessing the bending angle and comparing it to the simulation would be stronger proof. I suggest that the process of image modulation to get bending angles should be better described in methods part.

Response 11:

Improved in 3.3 High-temperature baking-induced deformation. The agreement and correlation between the simulation results and the actual induced deformation results are illustrated. (Lines 430-439)

The bending angle quantification is explained in 2.3 Quantification of bending angle(Lines 297-310)

Comments 12:

Figure 9 would be much easier to observe and analyse if presented in 2D.

Response 12:

Has been redrawn in 2D form (Figure 10, lines 447-449)

Comments 13:

In L393 authors state that model was printed while in the 2.1.2 there is named sample. I would suggest to make better denomination between simulated and printed samples. 

Response 13:

Thank you for the correction, here was a lack of clarity in my description, and a distinction has now been made between analogue and actual printed samples (lines 485-491)

Comments 14:

In L397 authors state "excellent bending" - there is nowhere defined what excellent bending would be. In the next paragraph (L399) authors conclude about the difference in heating, but there are no results supporting this statement. Similar to my statement earlier, in L422 there is also statement about excellent bending deformation.

Response 14:

Define what excellent bending is: The desired deformation is transverse deformation is the largest and most pronounced, and excess warpage at the edges of the model is the smallest and least pronounced(Lines 399-401)

The difference in heating is only a guess as to the cause of the warping of the model edges, here the description language is changed (lines 501-509) and a plan for exploring the warping is added to the "Future work" (lines 545-547).

Comments 15:

Conclusion part is generally to long. The aim of this paper was to determine COMSOL simulation of 3D printed objects. And the results and conclusions are more stressing the samples behaviour in surrounding of elevated temperature, this should be improved.

Response 15:

The structure of the conclusion section has been significantly modified to highlight the importance and significance of COMSOL in this study (4. Conclusion, line 480)

Comments 16:

Additionally, is there enough sample/models to evaluate the system for modelling? In my opinion having only three samples with same structure (fence like) should be improved. 

Response 16:

The aim of this paper is to provide a method that can cost-effectively obtain 3D printing condition parameters and is important to be able to simulate printed part variations while recording part variation data. This study will attempt to find such a method and provide new ideas for subsequent 3D printing related research. (Lines 72-76)

There is no suitable other will COMSOL for food 3D printing related research now, but I refined my conclusion section. The structure of the conclusion section has been significantly modified to highlight the importance and significance of COMSOL in this study (4. Conclusion, line 480)

Reviewer 2 Report

Comments and Suggestions for Authors

Overall the work submitted by authors is interesting. however, htere few comments that need revision. 

1. Regarding the introduction, your shoudl focus more on the current aim of your work. What the point of the particle and whihc is the hypothesis for that? This needs to be rewritten in more detail. 

2. Regarding the methodology, some areas needs clarification. For example in comsol you shoudl describe with detail the equations that you are applying and all the data for creating the model so readers coudl reproduce your work. In the current state it is difficutl to follow how did you reach to this point. 

3. The results make sense, but I am missing statisticla analysis, do you actually have statistical significna tdifference?

4. the discussion is poor, you should compare your results with those obtained by other authors. What is the real impact of your paper?

Comments on the Quality of English Language

Spelling and gramma errors shouls be corrected. 

Author Response

Dear Reviewer:

Your review is very detailed and pertinent, pinpointing areas of improvement in the manuscript. I feel that you must have reviewed my manuscript very carefully and had the idea of instructing a back like me, so you pinpointed my shortcomings.

Thank you very much for your work, I have carefully considered all the comments you have made and tried to revise them in accordance with your comments, I hope you will contact me if there are any shortcomings, and I will respond as soon as possible.

Thank you again for your guidance!

Sincerely

Author

Response to Reviewer's Comments

Summary

Thank you very much for taking the time to review the manuscript. I have responded to each of your comments below and have made the appropriate revisions in the resubmitted manuscript.

Comments 1:

Regarding the introduction, your shoudl focus more on the current aim of your work. What the point of the particle and whihc is the hypothesis for that? This needs to be rewritten in more detail.

Response 1:

The content and structure of the introductory section of the manuscript has been reorganized:

  1. in the first paragraph the application areas of AM are introduced before presenting the limitations of the current research on 3D printing of food products (lines 51-57) and an attempt to find ways to break these limitations (lines 72-76).
  2. focusing on the role and importance of COMSOL in this study, explaining the problems that can be solved using COMSOL, the kind of data and results that can be obtained, and highlighting the need to use COMSOL. (Lines 153-162)

Comments 2:

Regarding the methodology, some areas needs clarification. For example in comsol you shoudl describe with detail the equations that you are applying and all the data for creating the model so readers coudl reproduce your work. In the current state it is difficutl to follow how did you reach to this point. 

Response 2:

Data on the generalized Maxwell model are already available in COMSOL, and only a brief introduction to the Maxwell model is given here. Moreover, PET plastic film and starch gel materials are already available in the material library of COMSOL, so users only need to input some material properties (e.g., rheological properties, textural properties, etc.) of the starch gel. A description has been added to 2.1.2 (lines 242-247).

Comments 3:

The results make sense, but I am missing statisticla analysis, do you actually have statistical significna tdifference?

Response 3:

The object of the study was centred around three fence models of different widths, and three identical samples of each model were printed for parallel experiments (described in 3.3, lines 391-393) to obtain more reliable data.

Comments 4:

The discussion is poor, you should compare your results with those obtained by other authors. What is the real impact of your paper?

Response 4:

The aim of this paper is to provide a method that can cost-effectively obtain 3D printing condition parameters and is important to be able to simulate printed part variations while recording part variation data. This study will attempt to find such a method and provide new ideas for subsequent 3D printing related research. (Lines 72-76)

There is no suitable other will COMSOL for food 3D printing related research now, but I refined my conclusion section. The structure of the conclusion section has been significantly modified to highlight the importance and significance of COMSOL in this study (4. Conclusion, line 480)

Reviewer 3 Report

Comments and Suggestions for Authors In this manuscript, the authors simulated the starch gel printing and deformation process using COMSOL. Although the authors applied a simulation-experiment integrated method, the novelty of this work was not clear. It was unclear why the authors had to involve COMSOL into the study. In another word, why was COMSOL so important and necessary in this work. In addition, the authors simulated the deformation that might match the experimental results. But that's just the application of COMSOL, not the novelty of this study. it was unknown the unique contributions of this work.     Other major comments were listed as follows: 1. In the introduction, please focus on discussing the limitations of current papers that used COMSOL to simulate 3D printing. It's unclear the novelty of this work. 2. Table 1 was useless. If just the width was changed, it was unnecessary to include other constant parameters. 3. Font size in Figure 1 was too small. 4. Part of Table 2 was repeated with that in Table 1. 5. Figure 9 was weird. It's unnecessary to use 3D plots to make the figure which was difficult to read. Also, for all figures, please make sure the font size was consistent. 6. How was the consistence between the simulation and experimental results? How to determine whether the simulation was accurate or not? 7. Many typos and mistakes in the manuscript, such as citations in the abstract, typos, and "、".

Comments on the Quality of English Language

Lots of typos and mistakes in the manuscript. Please proofread before re-submission.

Author Response

Dear Reviewer:

Your review is very detailed and pertinent, pinpointing areas of improvement in the manuscript. I feel that you must have reviewed my manuscript very carefully and had the idea of instructing a back like me, so you pinpointed my shortcomings.

Thank you very much for your work, I have carefully considered all the comments you have made and tried to revise them in accordance with your comments, I hope you will contact me if there are any shortcomings, and I will respond as soon as possible.

Thank you again for your guidance!

Sincerely

Author

Response to Reviewer's Comments

Summary

Thank you very much for taking the time to review the manuscript. I have responded to each of your comments below and have made the appropriate revisions in the resubmitted manuscript.

 Comments 1:

In this manuscript, the authors simulated the starch gel printing and deformation process using COMSOL. Although the authors applied a simulation-experiment integrated method, the novelty of this work was not clear. It was unclear why the authors had to involve COMSOL into the study. In another word, why was COMSOL so important and necessary in this work. 

In addition, the authors simulated the deformation that might match the experimental results. But that's just the application of COMSOL, not the novelty of this study. it was unknown the unique contributions of this work. 

  1. In the introduction, please focus on discussing the limitations of current papers that used COMSOL to simulate 3D printing. It's unclear the novelty of this work. 

Response 1:

The content and structure of the introductory section of the manuscript has been reorganized:

  1. in the first paragraph the application areas of AM are introduced before presenting the limitations of the current research on 3D printing of food products (lines 51-57) and an attempt to find ways to break these limitations (lines 72-76).
  2. focusing on the role and importance of COMSOL in this study, explaining the problems that can be solved using COMSOL, the kind of data and results that can be obtained, and highlighting the need to use COMSOL. (Lines 153-162)

Comments 2:

Table 1 was useless. If just the width was changed, it was unnecessary to include other constant parameters.

Response 2:

The same data in the table is presented in the form of merged cells, and the different size information is displayed separately. (Lines 224)

Comments 3:

Font size in Figure 1 was too small.

Response 3:

The image has been redrawn so that the font in the image is as large as possible (lines 225)

Comments 4:

Part of Table 2 was repeated with that in Table 1. 

Response 4:

Personally, I think Table 1 is the actual print parameters and Table 2 is the model parameters in COMSOL, which may need to be redescribed again. Of course, this is just my opinion, and I am at your mercy as to how the tables are ultimately created. (lines 248)

Comments 5:

Figure 9 was weird. It's unnecessary to use 3D plots to make the figure which was difficult to read. Also, for all figures, please make sure the font size was consistent. 

Response 5:

It has been redrawn in 2D form (lines 447-449), and all the graphics have the same font size, just some of the images in the article have been scaled down.

Comments 6:

How was the consistence between the simulation and experimental results? How to determine whether the simulation was accurate or not? 

Response 6:

Improved in 3.3 High-temperature baking-induced deformation. The agreement and correlation between the simulation results and the actual induced deformation results are illustrated. (Lines 430-439)

The bending angle quantification is explained in 2.3 Quantification of bending angle (Lines 297-310)

Comments 7:

Many typos and mistakes in the manuscript, such as citations in the abstract, typos, and "、". 

Response 7:

Improvements have been made to statements throughout the text.

Round 2

Reviewer 1 Report

Comments and Suggestions for Authors

The article improved, but still I think that the conclusion part is too long. Significant part of it is more discussion than conclusion, where only the  key finding should be noted.

Comments on the Quality of English Language

English is improved, minor editing is suitable.

Author Response

Many thanks to you for taking the time to review my manuscript. The second round of revisions to the manuscript will now be explained:

For the problem of too much discussion in the conclusion section:

  1. transfer the discussion to the results section and revise it to blend in with the original content (lines 353-357, 429-445)
  2. and modify the real conclusion paragraph to emphasise the reliability of the COMSOL simulation.

Minor editing is suitable.:

I have also edited the English in the manuscript.

Reviewer 3 Report

Comments and Suggestions for Authors

The revised manuscript was significantly improved. I don't have more comments.

Comments on the Quality of English Language

English improvement was still needed.

Author Response

Many thanks to you for taking the time to review my manuscript. 

I have  edited the English in the manuscript.